# Cost Effective and Sustainable Test Methods to Investigate Vulnerabilities of EMP Attack on Existing Buildings Using Public Frequency Devices

**Chaeyeon Lim [1], Kyungryeung Min [2], Kukjoo Kim [1] and Young-Jun Park [1,*]**

1   Department of Civil Engineering and Environmental Sciences, Korea Military Academy, Seoul 01805, Korea; lcymail0210@gmail.com (C.L.); kukjoo.kim@mnd.go.kr (K.K.)
2   ICT Polytech Institute of Korea, 16-26 Sunam-ro Gwangju, Gyeonggi 12777, Korea; iolapleader@gmail.com
*   Correspondence: parky@mnd.go.kr; Tel.: +82-02-2197-2955

**Abstract:** High-power electromagnetic pulses are electromagnetic shock waves with strong energy, which can cause the destruction or malfunction of both electrical and electronic systems. The measurement of the electromagnetic wave shielding performance of buildings is effective for determining the level of EMP protection for each facility. However, it is extremely inefficient to practically measure the electromagnetic wave shielding performance using methods based on various standards for general buildings, in term of testing cost and time. Therefore, cost effective and sustainable test methods to investigate vulnerabilities of electromagnetic pulses attack on existing buildings using public frequency is proposed in this study. The study focuses on a simpler and more reliable method such as the use of broadcast or communication signals. From the results, it was concluded that the shielding performance can be measured approximately −45 dB using broadcast devices, −70 dB using walkie-talkie as simplified measurements of the electromagnetic-wave shielding performance. Therefore, if the test methods which are proposed in this study are allowed for preliminary investigation to find the vulnerabilities of existing buildings, cost and energy in the investigation can be reduced and it is expected to bring out frequency inspection and sustainable EMP protection performance of buildings.

**Keywords:** high-power electromagnetic pulses; electromagnetic wave shielding; public frequency; shielding effectiveness of building

## 1. Introduction

High-power electromagnetic pulses (EMPs) are electromagnetic shock waves with strong energy, which can cause the destruction or malfunction of both electrical and electronic systems. World powers including the United States and Russia have been steadily conducting studies on protection as well as weapon development using EMPs after the destruction and failure of various electronic devices by EMPs from nuclear explosions. The Republic of Korea military began to study the standard for EMP protection since the first nuclear test of North Korea in October 2006, and has been making significant efforts and conducting studies, including the establishment of the DMFC 2-20-30 standard for national defense and military facilities (design standards for electromagnetic wave protection facilities) suitable for the Korean environment [1,2].

Additionally, there is a risk of threatening the based system by intentionally generating EMPs in main information and communication facilities using electronic weapons. In other words, there is an increasing threat that can jam or destroy the equipment used, such as electronic bombs and high power electromagnetic wave generators other than nuclear weapons; currently, such devices are being miniaturized enough for portability by individuals because EMPs can be generated even with low power. Because such small-scale non-nuclear EMP equipment can be used anytime and anywhere, the risk of localized

terrorist threats is increasing, such as invading main information and communication facilities and deliberately generating EMPs to easily destroy electronic systems.

The defensive provisions currently in use in Korea have applied the protection method through the electromagnetic shielding room according to the standard (MIL-STD) established by the US military in the 1980s [3]. The standard is extremely strict because it is applied in military facilities and is configured to shield all facilities because temporary damage is not allowed owing to the nature of the military command system. However, saving measures such as partial protection or redundancy of only important equipment requiring protection are needed because such methods incur high costs for facility construction, maintenance, and repair.

However, in the recently constructed facilities, a method that continuously performs a function in response to a failure from EMPs can be applied. It is necessary to establish various defensive provisions by preparing and applying a system for the rapid failure recovery of target facilities through vulnerability analysis. Therefore, measuring the electromagnetic wave shielding performance of general buildings is effective for determining the level of EMP protection for each facility. However, it would be extremely inefficient to practically measure the electromagnetic wave shielding performance through methods based on various standards for general buildings in term of testing cost and time.

Therefore, cost effective and sustainable test methods to investigate vulnerabilities of EMP attack on existing buildings using public frequency are proposed in this study. The study focuses on a simpler and more reliable method such as the use of broadcast or communication signals considering that conventional measurements of electromagnetic wave shielding performance require significant measurement costs when targeting large scale buildings. This study is organized as follows:

1. A preliminary study was conducted to evaluate the measurement criteria for EMP shielding performance.
2. The public frequency for measuring the EMP shielding performance was considered.
3. The range of possible measurements of the EMP shielding performance by public frequency was determined through an experiment.

Since the simple measurement method developed in this study was not adopted as a standard, it can be used as a preliminary investigation to find the vulnerabilities of existing buildings. It is expected that the test methods using public frequency proposed in this study will be used in the future for the preliminary investigation to find the vulnerabilities where a precise measurement is required.

## 2. Materials and Methods

To establish an effective EMP protection facility, it is necessary to ensure that the main equipment and facilities inside the protection facility operate normally against EMP effects caused by nuclear explosions or EMP bombs. Particularly, the main equipment inside the protection facility should be protected by blocking or sufficiently attenuating the high voltage and current induced by the EMP effect. Generally, the electric field intensity of EMPs generated by a nuclear explosion is 50 kV/m to 65 kV/m [4]. EMPs can be blocked only when there is an electromagnetic wave shielding effectiveness of approximately 80 dB to 100 dB considering the radiating tolerance of the equipment to be protected [4]. However, the level of defensive provisions of this standard includes a safety margin and is an extremely strict standard that does not allow simple malfunctions (screen flickering, LED flickering, etc.). Existing buildings or environmental conditions themselves play a role of blocking EMPs in the case of EMP protection unlike the cyber protection field. If the system to be protected is located inside the facility, effective defensive provisions can be established by reflecting this and varying the level of protection for each location. Therefore, it is necessary to measure the electromagnetic wave shielding performance of general buildings. The high-altitude electromagnetic pulse(HEMP) protection for ground-based $C^4$I(Command, Control, Communications, Computer, and Intelligence) facilities performing critical, 4 time-urgent missions (MIL-STD-188-125-1) [3] of the US Department

of Defense, IEEE-STD-299 of The Institute of Electrical and Electronics Engineers [5], which is collectively referred to as a private standard, and Notice No. 2018-14 of National Radio Research Agency of Korea, which is the notice on the safety evaluation standards and methods for high power and leakage electromagnetic waves [6] etc., can be used to measure the electromagnetic-wave shielding performance of the facility. These three standards have different test methods such as the target frequency, placement of transmitting and receiving antenna, and unit test area, as shown in Table 1. Measuring the electromagnetic wave shielding performance of general buildings by applying these standards is effective in determining the level of EMP protection for each facility. However, it would be extremely inefficient to practically measure the electromagnetic wave shielding performance on all sides of a general building at equal intervals in term of testing cost and time.

**Table 1.** Comparison of shielding effectiveness test standards.

| Standards | MIL-STD-188-125-1 | | IEEE-STD-299.1 | | 2018-14 | |
|---|---|---|---|---|---|---|
| Publication | 2005 | | 2006 | | 2018 | |
| Field | High-altitude Electromagnetic Pulse | | Electromagnetic Compatibility | | Electromagnetic Compatibility | |
| | 10 kHz ~ 100 kHz | 20 test frequencies | 9 kHz ~ 300 MHz | a few test frequencies | 10 kHz ~ 100 kHz | 20 test frequencies |
| | 100 kHz ~ 1 MHz | 20 test frequencies | 300 MHz ~ 600 MHz | 1 test frequency | 100 kHz ~ 1 MHz | 20 test frequencies |
| | 1 MHz ~ 10 MHz | 40 test frequencies | 600 MHz ~ 1 GHz | 1 test frequency | 1 MHz ~ 10 MHz | 40 test frequencies |
| Frequency | 10 MHz ~ 100 MHz | 150 test frequencies | 1 ~ 2 GHz | 1 test frequency | 10 MHz ~ 100 MHz | 150 test frequencies |
| | 100 MHz ~ 1 GHz | 150 test frequencies | 2 ~ 4 GHz | 1 test frequency | 100 MHz ~ 1 GHz | 150 test frequencies |
| | | | 4 ~ 8 GHz | 1 test frequency | | |
| | | | 8 ~ 18 GHz | 1 test frequency | | |
| Placement of ransmitting and receiving antenna | • Loop/Bi-conical/LP Ant<br>• transmitting: 2.05 m<br>• receiving: 1.0 m | | • Loop Ant<br>• transmitting: 0.3 m<br>• receiving: 0.3 m<br>• Bi-conical/Dipole/Horn Ant<br>• transmitting: 1.7 m<br>• receiving: 0.3 m | | • Loop/Bi-conical/LP Ant.<br>• transmitting: 2.0 m<br>• receiving: 1.0 m | |
| Location of transmitting antenna | Outside of shielding facility | | Outside of shielding facility | | Outside of shielding facility | |
| Unit test area | 3.05 m × 3.05 m | | 2.6 m × 1.5 m | | 3 m × 3 m | |

## 3. Review of EMP Measurement Method Using Public Frequency Devices

### 3.1. Public Frequency

Public frequency includes various frequencies such as broadcast, communication, and walkie-talkie frequencies as shown in Table 2 [7]. The shielding effectiveness trend can be determined using these frequencies. For example, if the broadcast frequency is measured as −20 dBm on the outside of building and −50 dBm on the inside of building, it can be deduced that the building itself has 30 dB of electromagnetic shielding effectiveness.

**Table 2.** Usage and terms by frequency range.

| Frequency | Lower than 30 kHz | Lower than 300 kHz | Lower than 3 MHz | Lower than 30 MHz | Lower than 300 MHz | Lower than 3 GHz | Lower than 30 GHz | Lower than 300 GHz | Higher than 300 GHz |
|---|---|---|---|---|---|---|---|---|---|
| Term | V.L.F. [1] | L.F. [2] | M.F. [3] | S.F. [4] | V.H.F. [5] | U.H.F. [6] | | Microwave | |
| Usage | Marine communication | GCA [7] | AM radio | Shortwave broadcasting, Radio amateurs | FM radio, TV broadcasting | Mobile, TV broadcasting | Satellite communication | Space communication | Radio astronomy |

[1] Very low frequency. [2] Low frequency. [3] Medium frequency. [4] Short frequency. [5] Very high frequency. [6] Ultra high frequency. [7] Ground controlled approach.

Although the general shielding effectiveness test can be conducted in the operating range of approximately 100 to 120 dB using a power amplifier and high gain antenna, only a low level of shielding effectiveness can be measured because the public frequency is tested in the operating range of approximately 30 to 50 dB. However, it has a sufficient operating area for trend prediction because the shielding effectiveness of reinforced concrete buildings is approximately 10 to 30 dB [8].

Radio waves such as military communications, satellite, radio, TV, and walkie-talkies are widely used in daily life. A simplified test was conducted to evaluate the degree of transmission of these public frequencies through general buildings. The test was conducted in an office building of the Korea Radio Promotion Association (RAPA). The environment noises in various environments such as rooftop as Figure 1a, outside on the ground as Figure 1b, inside of building nearby window as Figure 1c, inside of building nearby concrete wall as Figure 1d, basement floor as Figure 1e, and inside of electromagnetic shielding room as Figure 1f were measured. The frequency in the 300 MHz to 3 GHz band was measured using a log periodic vertical antenna and a mobile spectrum analyzer. The Max Hold function was used to measure the maximum instantaneous signal of the data.

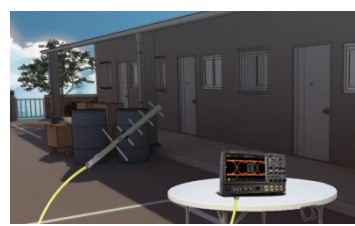

(**a**) Outside(rooftop)

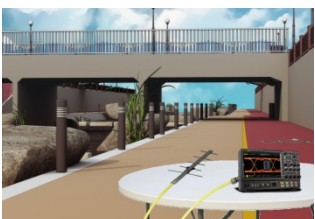

(**b**) Outside(ground)

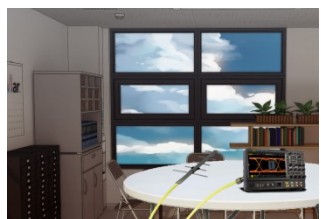

(**c**) Inside of building (nearby window)

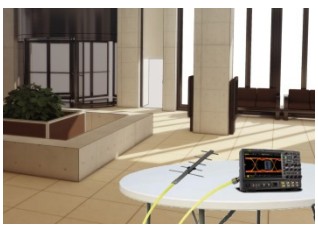

(**d**) Inside of building (nearby concrete wall)

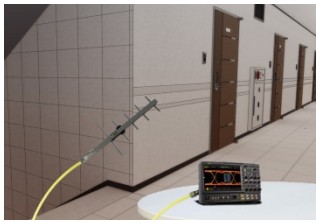

(**e**) Basement floor

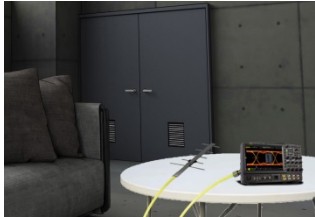

(**f**) Electromagnetic shield room

**Figure 1.** Test places for environment noise measurement.

As the result, the maximum environment noises of −27.32 dBm were measured on the rooftop as Figure 2a, −18.81 dBm on the outside on the ground as Figure 2b, −23.13 dBm on inside of building nearby window as Figure 2c, −48.18 dBm on inside of building nearby concrete wall as Figure 2d, −34.53 dBm on the basement floor as Figure 2e, and −56.63 dBm on inside of electromagnetic shielding room as Figure 2f. This result shows that the electromagnetic wave shielding effectiveness of reinforced concrete building is approximately −30 dBm in the 880 MHz band.

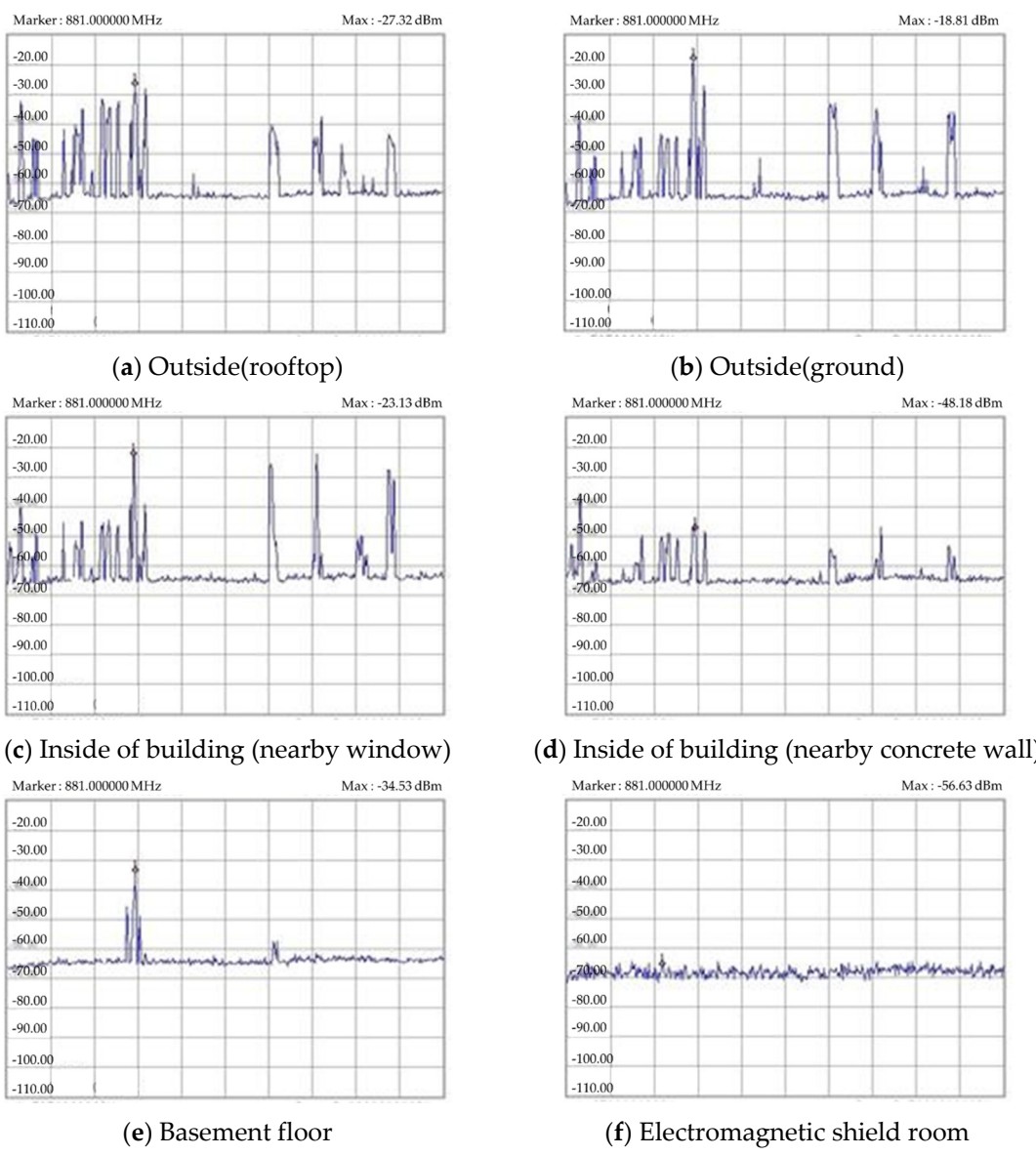

**Figure 2.** Results of the environment noise measurement.

### 3.2. Characteristics and Measurement Methods Based on Public Frequencies

3.2.1. Broadcast Frequency (AM, FM, TV, etc.)

The broadcast frequency has the longest history alongside the communication frequency and is the most familiar frequency to the public. Broadcast frequencies can be mainly divided into radio (AM, FM) and TV (VHF, UHF). First, in the case of the radio, it is divided into amplitude modulation (AM) and frequency modulation (FM), in which AM uses a frequency between 535 kHz and 1605 kHz and FM uses a frequency band between 88 MHz and 108 MHz [7]. In the case of the TV, it is mainly divided into VHF and UHF,

in which VHF is used for military, terrestrial digital multimedia broadcasting, and other purposes, and is distributed in the frequency bands of 54 MHz to 88 MHz and 174 MHz to 216 MHz, whereas UHF is used for terrestrial digital TV and new services in the frequency bands of 470 MHz to 698 MHz and 698 MHz to 806 MHz. However, a frequency signal should be received to obtain information or videos using all frequencies. This signal has a low reception rate in places with high environmental noise in the frequency band, making reception difficult in a place where the transmission signal is weak because of distance or a large number of jamming factors. Therefore, there is a standard for the electric intensity of broadcasting signals for all broadcasting areas as shown in Table 3 [9].

**Table 3.** Electric intensity of broadcasting area classified by type of broadcasting station.

| Type of Broadcasting Station | | Electric Intensity of Broadcasting Area (dBμV/m) | | |
| --- | --- | --- | --- | --- |
| | | High-Noise Area | Mid-Noise Area | Low-Noise Area |
| Standard broadcasting | | 77 | 74 | 71 |
| Frequency modulation broadcasting | | 70 | 60 | 48 |
| Digital terrestrial television broadcasting | Low VHF | 28 | | |
| | High VHF | 36 | | |
| | UHF | 41 | | |
| Ultra-definition television broadcasting | Low VHF | 38 | | |
| | High VHF | 40 | | |
| | UHF | 45 | | |
| Mobile multimedia broadcasting | | 45 | | |

Broadcast frequency signals of the above intensity can be received even inside buildings without much attenuation. If there is no environmental noise overlapping the broadcast frequency inside the building, it can be sufficiently applied as a simplified measurement method of the electromagnetic wave shielding performance of the building. The electromagnetic wave shielding performance measurement method using a broadcast frequency can select a different broadcast frequency band when reception is disabled because of a large amount of environmental noise or a weak transmission signal.

3.2.2. Walkie-Talkies

Walkie-talkies that can be used by the general public are mainly divided into three types: the walkie-talkie, radio agency, and amateur radio station. Among them, the walkie-talkie is the easiest and most accessible to the general public. The walkie-talkie is divided into walkie-talkie type 1 (citizens band radio service) and walkie-talkie type 2 (family radio service) as shown in Table 4 [10]. First, the frequency band of the Walkie-talkie type 1 uses a short wave of 27 MHz, and the average reception distance is 1 km to 3 km without installing a 1.5 m class large antenna. Walkie-talkie type 2 uses a frequency band of 448 MHz and it is difficult for the reception distance to exceed 3 km because it depends on the direct microwave. Additionally, because walkie-talkie type 1 has an output limit of 3 W or below in the short wave, and Walkie-talkie type 2 has an output limit of 0.5 W or below in the microwave, it is regulated at the national level to ensure that the effective transmission output radiated from the transmitting antenna cannot be amplified arbitrarily [10].

The electromagnetic wave shielding performance of a building can be measured using a walkie-talkie. This method can easily distinguish environmental noise because the frequency band is designated. It is used to obtain measurements by reducing the influence of environmental noise because it is divided into 15 to 40 various channels and can measure electromagnetic wave shielding performance in a desired place because it is portable.

**Table 4.** Type and attribute of walkie-talkie.

| Type | Walkie-Talkie Type 1 (Citizens Band Radio Service) | Walkie-Talkie Type 2 (Family Radio Service) |
|---|---|---|
| Frequency(type) | 27 MHz (short wave) | 448 MHz (microwave) |
| Modulation system | AM, FM | FM |
| Acceptance output | 3 W | 0.5 W |
| Number of channels | 40 | 15 |
| Allowed type | Portable, Car mounted, Fixed | Portable |
| Modes of transmission | Simplex | Simplex, duplex operation |

### 3.2.3. Mobile Phone Frequency

SK telecom, LGU plus, and KT which are mobile phone operators in Korea, occupy 49%, 30%, and 21%, respectively, of the Korean mobile communication market. A considerably wide frequency band of 728 MHz to 2690 MHz is distributed and used for mobile phones in Korea. Three telecommunication companies provide services using the different frequency bands allocated within the frequency band. Each company uses different frequency bands according to the communication speed they provide, creating competition for the allocation of good bands where radio frequency is not interfered with [7]. Smartphones, which are commonly used by the public, have a function that shows the strength of the signal being received numerically in the device itself. The strength of the received signal can be measured from approximately −50 dB at a relatively close distance from the base station, to approximately −130 dB in areas where the phone function does not work well or there is a radio shadow area. If a repeater is installed in the building for communication transmission and reception functions inside the building and underground, similar strength of signal with the outside of the building can be received [7]. Using these characteristics, the shielding performance of the building and that of the shielding room can be easily measured in cases where there is a shielding room inside the building.

### 3.2.4. Noise Emitter

The noise emitter is widely used for the comparison test or verification of measurement equipment in the electromagnetic compatibility (EMC) test and measurement. This equipment transmits signals of more than certain strength in a wide frequency band and each product has various functions. Although the noise emitter can only emit an output at a specific frequency, it generates and uses a signal in a certain frequency section in most cases. The desired frequency band and signal strength can be selected after checking the function of the product presented by the equipment seller.

### 3.2.5. ISM (Industrial Scientific and Medical) Frequency Band and Short-Range Wireless Devices

The ISM band is an unlicensed band intended for low-data rate communications and extends from 902 to 928 MHz mainly used for wireless networks such as IEE 802.11a/b/g/n. By definition, any transceiver operating in an unlicensed band does not require that the end user obtain a government permit to use the device. However, the device itself must be certified by the governing authority in the operating country. Due to its unlicensed nature, the 915-MHz U.S. ISM band is popular for establishing wireless data links with short-range wireless devices. As the popularity grows, the number of devices operating in this band increases. This situation can create a congested frequency spectrum, and this congestion manifests itself in interference from other devices operating in the band, which will degrade the performance of the intended link [11].

Because of the attribute of ISM frequency band such as unlicensed band and interference, ISM frequency band is not suitable for measuring the shielding effectiveness. Also, the other short-range wireless devices have similar attributes to the ISM frequency band.

Therefore, ISM frequency band and short-range wireless device is excluded from the scope of this study.

### 3.2.6. Other Test Methods

The shielding performance of buildings can be measured by continuous wave (CW) transmission and reception if there is equipment for measuring electromagnetic wave shielding performance such as the signal generator, spectrum analyzer, transmission/ reception antenna, and power amplifier. The measurable frequency band may vary depending on the performance of each of the measurement equipment. The desired representative frequency band can be selected within the measurable frequency band of the equipment, and the operating area of the equipment can be determined to measure the electromagnetic wave shielding performance if it is difficult to secure the measurement distance between the transmitting and receiving antennas and that of the reference value required by the standard, owing to the characteristics of the measured environment. First, the reference value measurement method determines the reference value by installing the transmitting/receiving antenna, spectrum analyzer, and power amplifier suitable for the selected frequency band at the furthest distance in a straight line from the inside of the building, and then transmitting and receiving the CW waveform. In the case of general reinforced concrete buildings, because the frequency attenuation characteristic increases as the measurement frequency band decreases, the lowest value between 30 MHz to 200 MHz can be applied equally to the band below 30 MHz [12]. Finally, the shielding performance of the building can be measured by transmitting the signal after placing the transmitting antenna at the center of the building and calculating the difference between the first measured reference value and the received signal value measured after installing the receiving antenna on the outer wall of the building.

## 4. EMP Shielding Effectiveness Measurement Experiment through Public Frequency

*4.1. Measurement of Electromagnetic-Wave Shielding Performance Using Broadcast and Communication Signals*

### 4.1.1. Installation and Scheme of Measurement

The signal strength difference between receiving signals on the inside and outside of the shielding room was measured to verify that the broadcast and communication signals could be used for simplified measurement methods of EMP shielding effectiveness.

Figure 3 shows the shielding room designed for shielding effectiveness test in this study. The frequency band of the spectrum analyzer for the measurement of signal strength was set between 30 MHz and 1.6 GHz which is similar frequency of the band of EMP, and set to SPAN 10 MHz, the coupling parameter RBW (Resolution Bandwidth) was set to 200 kHz, VBW (Video Bandwidth) was 500 kHz, and SWT (Sweep time in seconds) was 2.5 ms. The antenna was conducted by orientation towards to the opposite direction from the shielding room when measuring the outside of the shielding room, and towards to outside at the center of the shielding room when measuring the inside of the shielding room.

### 4.1.2. Analysis of Experiment Result

The maximum signal level for FM broadcasting was 91.53 MHz of −74 dBm and was measured as −63 dBm inside the shielding room, showing a difference of approximately 11 dB. The maximum signal level of terrestrial digital TV broadcasting was −67 dBm and the frequency in the 183 MHz band was the strongest. Further, the noise level inside the shielding room was measured as −72 dBm, indicating that there was a margin of approximately 5 dB. The strongest signal was the 880 MHz mobile communication frequency, which was measured as approximately −30 dBm from the outside and −75 dBm from the inside, resulting in a difference of approximately 45 dB as shown in Figure 4. This is believed to be a sufficient value to measure the electromagnetic wave shielding performance of general buildings.

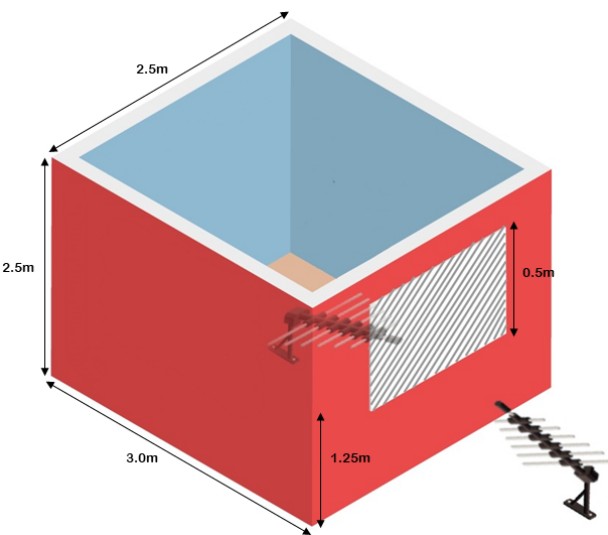

**Figure 3.** Shielding room setup.

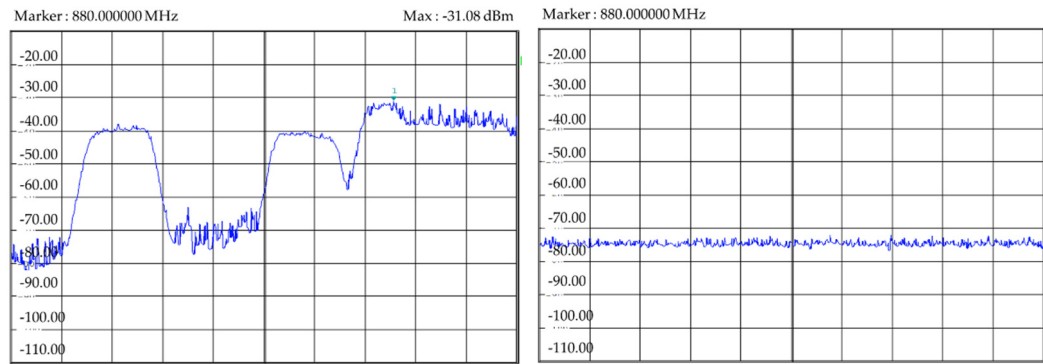

(**a**) outside of electromagnetic shielding room　　　　(**b**) inside of electromagnetic shielding room

**Figure 4.** Noise level of mobile communication frequency.

### *4.2. Measurement of Electromagnetic-Wave Shielding Performance Using Walkie-Talkie Signal*

#### 4.2.1. Installation and Scheme of Measurement

A test was conducted to check the signal level using a log periodic antenna in the radio wave of a walkie-talkie and determine the difference when measured indoors. A method of receiving walkie-talkie signals from outside the shielding room and receiving signals of the same frequency band inside the shielding room was used as a test method to evaluate the difference in shielding performance. The operating area of the measuring equipment was examined by measuring the walkie-talkie signal reception level outside the shielding room and the noise level in the same frequency band inside the shielding room. The walkie-talkie for business in the 400 MHz band was used for the walkie-talkie, and channel 15 with the least effect of environmental noise was used. For the spectrum analyzer settings, it was measured in two modes: RBW 500 Hz/(10 kHz), considering the operating area and measurement time of the measuring equipment and frequency band of 448.92 MHz, and the coupling parameter was set to 10 kHz span, VBW 2 kHz, and SWT 40 ms (Coupling Parameter 100 kHz span, VBW 30 kHz, and SWT 5 ms). The maximum signal (Max Hold) received from the walkie-talkie was measured at 1 m, 2 m, and 3 m from the antenna

outside the shielding room, and the noise level (Max Hold) was measured in the same frequency band inside the shielding room in similar receiver settings.

### 4.2.2. Analysis of Experiment Result

The measurement result was −3.47 dBm when the distance between the walkie-talkie and antenna was 1 m in the RBW 10 kHz mode, −10.0 dBm at 2 m, and −12.98 dBm at 3 m as shown in Table 5. Shielding of 67.76 dB, 61.23 dB, and 58.25 dB, respectively, was measured considering the noise level of −71.23 dBm. Noise levels of −5.25 dBm, −12.67 dBm, and −12.87 dBm were measured when the distance between the walkie-talkie and antenna was 1 m, 2 m, and 3 m, respectively, in the RBW 500 Hz mode. Additionally, shielding of 78.32 dB, 70.90 dB, and 70.70 dB, respectively, was also measured considering the noise level of –83.57 dBm. It was found that the maximum shielding performance was 78 dB when using the walkie-talkie. It is believed that the abnormality of the shielding performance can be checked by only transmitting and receiving walkie-talkies without a spectrum analyzer if the minimum reception levels for the walkie-talkie voice calls are identified.

**Table 5.** Measurement results of walkie-talkie signal.

| RBW Setting | Noise Level (dBm) | | | |
| --- | --- | --- | --- | --- |
| | 1 m Distance from Antenna | 2 m Distance from Antenna | 3 m Distance from Antenna | In Shielding Room |
| 10 kHz | −3.47 | −10 | −12.98 | −71.23 |
| 500 Hz | −5.25 | −12.67 | −12.87 | −83.57 |

### 5. Discussion and Conclusions

There are practical limitations for non-EMP-related experts to fully apply the method recommended in the standard for measuring and analyzing EMP vulnerability of general buildings. Moreover, it would be extremely inefficient to practically measure the electromagnetic wave shielding performance through methods based on various standards for general buildings in term of testing cost and time. Therefore, cost effective and sustainable test methods to investigate vulnerabilities of EMP attack on existing buildings using public frequency are proposed in this study.

First, the shielding performance could be measured up to approximately 45 dB by using broadcast frequencies (AM, FM, TV, etc.) in a test. The broadcast frequency has the advantage of a wide selection of the desired frequency band because a wide frequency band is used although the reception state of the signal may be different depending on the shadow area where the frequency reception signal is weak.

Second, the highest shielding performance of up to 78 dB was measured from tests in the 400 MHz band using a walkie-talkie signal, which is a representative communication device. In the case of the walkie-talkie, the range of frequency selection is narrow because the usable frequency band is fixed. However, it has the advantage that it can measure a higher shielding performance than that of the broadcast signal because the strength of the transmit and receive signal of the walkie-talkie is strong and portable.

Third, because the method of measuring the shielding performance using a mobile phone frequency and noise emitter uses a considerably wide frequency band, the shielding performance over a wide frequency band can be measured. Also, it has the advantage that it can be measured anywhere with signal amplification using many repeaters, and it is portable. Although measurements using a mobile phone frequency and noise emitter were not performed in this study, the study enabled a wider range of measurements than the method using a broadcast frequency or a walkie-talkie signal depending on the characteristics of the device.

It was concluded that up to approximately 70 dB can be measured using public frequency devices such as broadcasting, communication, and walkie-talkie devices in the

simplified measurement method of electromagnetic wave shielding performance described above. Since the simple measurement method proposed in this study was not adopted as a standard, it can be used as a preliminary investigation to find the vulnerabilities of existing buildings.

For example, according to Article 98-2 of the Enforcement Decree of Radio Waves Act of Korea, a person who files a request for the safety assessment of radioactive protection efficiency need to pay approximately $268 as a fee per measuring point [13]. Therefore, the client who request to assessment of conductivity protection efficiency for 100 measuring point in accordance with Korean code, need to pay approximately total $26,800. If the test methods which is proposed in this study, are allowed for preliminary investigation to find the vulnerabilities of existing buildings, the client does not need to request to assessment of conductivity protection efficiency for the all the measuring point. As a result, the cost and energy reduction in the investigation of electromagnetic wave shielding effectiveness are expected to bring out frequency inspection and sustainable EMP protection performance of buildings.

**Author Contributions:** Conceptualization, K.M. and Y.-J.P.; methodology, K.M. and K.K.; software, K.M.; validation, C.L., K.M., K.K. and Y.-J.P.; formal analysis, K.M. and K.K.; investigation, C.L. and Y.-J.P.; resources, K.M. and K.K.; data curation, C.L. and Y.-J.P.; writing—original draft preparation, C.L. and K.K.; writing—review and editing, C.L. and Y.-J.P.; visualization, C.L. and K.K.; supervision, Y.-J.P.; project administration, K.K. and Y.-J.P.; funding acquisition, Y.-J.P. All authors have read and agreed to the published version of the manuscript.

**Funding:** This research was supported by a grant (18SCIP-B146646-01) from the Korea Agency for Infrastructure Technology Advancement.

**Institutional Review Board Statement:** Not applicable.

**Informed Consent Statement:** Informed consent was obtained from all subjects involved in the study.

**Data Availability Statement:** Data sharing is not applicable to this article.

**Acknowledgments:** This work was supported by research fund of the Korea Agency for Infrastructure Technology Advancement. The ROKA Nuclear WMD Protection Research Center at Korea Military Academy is gratefully acknowledged for providing the support that made this study possible.

**Conflicts of Interest:** The authors declare no conflict of interest. The funders had no role in the design of the study; in the collection, analyses, or interpretation of data; in the writing of the manuscript, or in the decision to publish the results.

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
