# Peer review of "Cost Effective and Sustainable Test Methods to Investigate Vulnerabilities of EMP Attack on Existing Buildings Using Public Frequency Devices"

_sustainability, doi:10.3390/su13020570_

Round 1

Reviewer 1 Report

Dear authors,

Congratulations for this very interesting and useful paper, in which the measurement methods of the electromagnetic-wave shielding performance of buildings using public frequency are proposed.

Regarding the modality in which this paper was written, I have some suggestions:

Title: to include the “buildings” in the title, for example: “Simplified methods for measuring the electromagnetic-wave shielding performance of buildings using  public frequency”

Abstract: The summary should detail the results obtained in the experimental study done.

Keywords: To complete the keywords with the word “building performance”

Other suggestions:

  1. p. 81 - to add: the electric field intensity of EMPs
  2. p. 82 - The phrase “… the electric field intensity of EMPs generated by a nuclear explosion is 50 kV/m or 65 kV/m” should be confirmed detailed with specific references (In fact, some HPEM simulators generate electric fields of intensity of 50 kV/m or 60 kV/m).
  3. p. 96-97 – the three methods… – which methods (detail)!
  4. p. 112 – Figure 1 should be redesigned, and a suggestive title to be given.
  5. p. 119 – To add and other references regarding to the shielding effect on concrete of the building (see IEEE Transactions on Electromagnetic Compatibility papers).
  6. p. 122-132 – It is not clear: are the experiments of the authors or are the cited experiments? Give the reference if they are not your experiments.
  7. p. 135 – Give a suggestive title of the figure 2.
  8. p. 134 – Figure 3 should be introduced, and some analyses of the results should be done.
  9. 135 – The title (and content) of the section 3.2, and subsections 3.2.1 – 3.2.5, should be enlarged, to include not only classification of the public signals, but also the analysis of the possibility of use the public frequencies for measuring the EMP shielding effect. Example: Measurement methods based on public frequencies (for 3.2), or , Measurement method with Broadcast frequency, etc.
  10. p.222 – The section 4.1 should be split in sub-sections (with description of installation and scheme of measurement; measurements; data processing and analysis)
  11. p. 230 – Detail what are RBW, VBW, RWT. A figure with the measurement scheme is better to be introduced. Give details on the shielding room.
  12. p.243 - Title of Figure 4 – should be in connection of the broadcast signals or wave forms of these signals
  13. p. 245 – The section 4.2 should have the same structure as for 4.1.
  14. p. 273-303 – The Discussion and Conclusions should be bête systemized, and a comparative analysis should be developed. Some conclusions should be included in abstract.

Author Response

Response to Reviewers’ Comments

Manuscript ID.: sustainability-1045667

Title: Simplified Measurement Methods of electromagnetic-wave shielding performance using public Frequency

The authors would like to first thank the editor who allowed us opportunities to revise and resubmit the paper. We also sincerely appreciate the anonymous reviewer who provided thorough reviews and valuable comments to help us improve the manuscript. We strongly believe that in the revision we have fully addressed all the reviewer’s comments and concerns and carefully revised the manuscript based on the feedback we have received. Please see the followings below responding to reviewer’s comments and attached file which has highlight with red color on revised part.

Reviewer 1

Congratulations for this very interesting and useful paper, in which the measurement methods of the electromagnetic-wave shielding performance of buildings using public frequency are proposed.

Regarding the modality in which this paper was written, I have some suggestions:

Comment 1. Title: to include the “buildings” in the title, for example: “Simplified methods for measuring the electromagnetic-wave shielding performance of buildings using public frequency”

Response.

We changed the title of paper in revised manuscript as follows.

Original.

Simplified Measurement Methods of electromagnetic-wave shielding performance using public Frequency

Revised.

Simple Measurement Methods of Electromagnetic-wave Shielding Performance for Existing Buildings using Public Frequency Devices

Comment 2. Abstract: The summary should detail the results obtained in the experimental study done

Response.

We revised the abstract as follows.

Original.

High-power electromagnetic pulses (EMPs) are electromagnetic shock waves with strong energy, which can cause the destruction or malfunction of both electrical and electronic systems. Currently, a method that enables continuous functioning in response to failures from EMPs can be applied to facilities that are under construction. Various defensive provisions need to be established by preparing and applying a system for rapid failure recovery of the target facilities through vulnerability analysis. Therefore, the measurement of the electromagnetic-wave shielding performance of buildings is effective for determining the level of EMP protection for each facility. However, it is extremely inefficient to practically measure the electromagnetic-wave shielding performance using methods based on various standards for general buildings, owing to high economic and time burdens. Therefore, a method of measuring the electromagnetic-wave shielding performance of existing buildings using public frequency is proposed in this study. The study focuses on a simpler and more reliable method such as the use of broadcast or communication signals. From the results, it was concluded that the performance can be measured up to approximately 70 dB using devices that use public frequency such as broadcast, communication and walkie-talkie signals as simplified measurements of the electromagnetic-wave shielding performance.

Revised.

High-power electromagnetic pulses (EMPs) are electromagnetic shock waves with strong energy, which can cause the destruction or malfunction of both electrical and electronic systems. The measurement of the electromagnetic-wave shielding performance of buildings is effective for determining the level of EMP protection for each facility. However, it is extremely inefficient to practically measure the electromagnetic-wave shielding performance using methods based on various standards for general buildings, in term of testing cost and time. Therefore, a method of measuring the electromagnetic-wave shielding performance of existing buildings using public frequency is proposed in this study. The study focuses on a simpler and more reliable method such as the use of broadcast or communication signals. From the results, it was concluded that the shielding performance can be measured approximately -45 dB using broadcast devices, -70 dB using walkie-talkie as simplified measurements of the electromagnetic-wave shielding performance. However, the simple measurement method developed in this study has not been adopted as a standard, it can be used as a preliminary investigation to determine the vulnerabilities of existing buildings and whether to perform precise measurements in the future.

Comment 3. Keywords: To complete the keywords with the word “building performance”

Response.

We revised the keyword as follows.

Original.

Keywords: high-power electromagnetic pulses; electromagnetic-wave shielding; public frequency

Revised.

Keywords: high-power electromagnetic pulses; electromagnetic-wave shielding; public frequency, shielding effectiveness of building

Comment 4. p. 81 - to add: the electric field intensity of EMPs

Comment 5. The phrase “… the electric field intensity of EMPs generated by a nuclear explosion is 50 kV/m or 65 kV/m” should be confirmed detailed with specific references (In fact, some HPEM simulators generate electric fields of intensity of 50 kV/m or 60 kV/m).

Response.

About comment 4, we revised as follows.

About comment 5, there was some mistakes in translation and editing process. The phrase was referenced from reference number 4. So. we revised as follows.

Original. (line 79~81)

 Particularly, the main equipment inside the protection facility should be protected by blocking or sufficiently attenuating the high voltage and current induced by the EMP effect [4]. Generally, the electric intensity of EMPs generated by a nuclear explosion is 50 kV/m or 65 kV/m.

Revised. (line 86~89)

Particularly, the main equipment inside the protection facility should be protected by blocking or sufficiently attenuating the high voltage and current induced by the EMP effect. Generally, the electric field intensity of EMPs generated by a nuclear explosion is 50 kV/m to 65 kV/m [4].

Comment 6. p. 96-97 – the three methods… – which methods (detail)!

Response.

There was some mistakes in translation process. That means the standards(MIL-STD-188-125-1, IEEE-STD-299 , Notice No. 2018-14 of National Radio Research Agency of Korea). So. we revised as follows.

Original. (line 96-97)

 These three measurement methods have different test methods such as the target frequency, placement of transmitting and receiving antenna, and unit test area, as shown in Table 1.

Revised. (line 104)

These three standards above have different test methods such as the target frequency, placement of transmitting and receiving antenna, and unit test area, as shown in Table 1.

Comment 7. p. 112 – Figure 1 should be redesigned, and a suggestive title to be given.

Response.

We revised the figure 1 from figure to table and the title as follows.

Original.

Figure 1. Usage by frequency

Revised.

Table 2. Usage and terms by frequency range

Frequency

Lower than  30 kHz

Lower than  300 kHz

Lower than  3 MHz

Lower than  30 MHz

Lower than  300 MHz

Lower than   3 GHz

Lower than  30 GHz

Lower than  300 GHz

Higher than  300 GHz

Term

V.L.F.1

L.F. 2

M.F. 3

S.F. 4

V.H.F. 5

U.H.F. 6

Microwave

Usage

Marine communication

GCA7

AM radio

Shortwave broadcasting, Radio amateurs

FM radio, TV broadcasting

Mobile, TV broadcasting

Satellite communication

Space communication

Radio astronomy

Comment 8. p. 119 – To add and other references regarding to the shielding effect on concrete of the building (see IEEE Transactions on Electromagnetic Compatibility papers).

Comment 9. p. 122-132 – It is not clear: are the experiments of the authors or are the cited experiments? Give the reference if they are not your experiments.

Response.

About the comment 8, we add a reference for that sentence.

About the comment 9, we did the experiments in RAPA(the Korea Radio Promotion Association).

Also we revised the section 3.1 for better understanding as follow

Original. (line 115~131)

Although the general shielding effect test can be conducted in the operating range of approximately 100 to 120 dB using a power amplifier and high gain antenna, only a low level of shielding effect can be measured because the public frequency is tested in the operating range of approximately 30 to 50 dB. However, it has a sufficient operating area for trend prediction because the shielding effect of reinforced concrete buildings is approximately 10 to 30 dB. Radio waves such as military communications, satellite, radio, TV, and walkie-talkies are widely used in daily life. A simplified test was conducted to evaluate the degree of transmission of these public frequencies through general buildings. The test was conducted in an office building of the Korea Radio Promotion Association (RAPA) as shown in Figure 2, where the environment noises in various environments such as above ground, underground, and rooms surrounded by reinforced concrete were measured. The frequency in the 300 MHz to 3 GHz band was measured using a log periodic vertical antenna and a mobile spectrum analyzer. The Max Hold function was used to measure the maximum instantaneous signal of the data. From the measurements, a maximum of -18.81 dBm in the 880 MHz band was measured under above-ground conditions and -48.18 dBm was measured in the room surrounded by reinforced concrete. From the results, d) the inside of the building (with concrete wall) had a shielding effect of approximately 30 dBm in the 880 MHz band as shown in Figure 3, and the electromagnetic wave shielding tendency can be analyzed.

Revised. (line 125~144)

Although the general shielding effectiveness test can be conducted in the operating range of approximately 100 to 120 dB using a power amplifier and high gain antenna, only a low level of shielding effectiveness can be measured because the public frequency is tested in the operating range of approximately 30 to 50 dB. However, it has a sufficient operating area for trend prediction because the shielding effectiveness of reinforced concrete buildings is approximately 10 to 30 dB [8].

Radio waves such as military communications, satellite, radio, TV, and walkie-talkies are widely used in daily life. A simplified test was conducted to evaluate the degree of transmission of these public frequencies through general buildings. The test was conducted in an office building of the Korea Radio Promotion Association (RAPA). The environment noises in various environments such as rooftop (a), outside on the ground (b), inside of building nearby window (c), inside of building nearby concrete wall (d), basement floor (e), and inside of electromagnetic shielding room (f) were measured as shown in Figure 1. The frequency in the 300 MHz to 3 GHz band was measured using a log periodic vertical antenna and a mobile spectrum analyzer. The Max Hold function was used to measure the maximum instantaneous signal of the data.

As the result, the maximum environment noises of -27.32 dBm were measured on the rooftop(a), -18.81 dBm on the outside on the ground(b), -23.13 dBm on inside of building nearby window (c), -48.18 dBm on inside of building nearby concrete wall (d), -34.53 dBm on the basement floor (e), and -56.63 dBm on inside of electromagnetic shielding room (f) as shown in the Figure 2. This result shows that the electromagnetic wave shielding effectiveness of reinforced concrete building is approximately -30 dBm in the 880 MHz band.

Comment 10. p. 135 – Give a suggestive title of the figure 2.

Response.

We changed the title of figure 2 as follows.

Original.

Figure 2. Measurement of public frequency.

Revised.

Figure 1. Test places for environment noise measurement

Comment 11. p. 134 – Figure 3 should be introduced, and some analyses of the results should be done.

Response.

Because  we revised the figure 1 from figure to table, figure 3 in original manuscript is changed to figure 2. And we add some description about figure 2 in line 139~144

Comment 12. p.135 – The title (and content) of the section 3.2, and subsections 3.2.1 – 3.2.5, should be enlarged, to include not only classification of the public signals, but also the analysis of the possibility of use the public frequencies for measuring the EMP shielding effect. Example: Measurement methods based on public frequencies (for 3.2), or , Measurement method with Broadcast frequency, etc.

Response.

We changed the title 3.2 as follow.

Original.

3.2. Characteristics of each public frequecy type

Revised.

3.2. Characteristics and measurement methods based on public frequencies

Comment 13. p.222 – The section 4.1 should be split in sub-sections (with description of installation and scheme of measurement; measurements; data processing and analysis)

Response.

We changed the title and contents of section 4.1 in many part.

Please check the revised manuscript 

Comment 14. p. 230 – Detail what are RBW, VBW, RWT. A figure with the measurement scheme is better to be introduced. Give details on the shielding room.

Response.

The detail of RBW, VBW and SWT was in line 254, 256, and 257 even it was mentioned first in line 230. So, the detail is moved to the location where was mentioned first (line 243~244 in revised manuscript)

About the shielding room, we add figure 3 for more detail of it.

Comment 15. p.243 - Title of Figure 4 – should be in connection of the broadcast signals or wave forms of these signals

Response.

We changed the title of figure 4 as follows.

Original.

Figure 4. measurement result of electromagnetic-wave shielding performance (broadcast signal).

Revised.

Figure 4. Noise level of mobile communication frequency 

Comment 16. p. 245 – The section 4.2 should have the same structure as for 4.1.

Response.

We changed the structure of section 4.2 as same with 4.1

Comment 17. p. 273-303 – The Discussion and Conclusions should be bête systemized, and a comparative analysis should be developed. Some conclusions should be included in abstract.

Response.

We revised the ‘Abstract’ and ‘Discussion and Conclusions”

Please check the revised manuscript 

Reviewer 2 Report

The paper concerns measurement of shielding performance of building walls against electromagnetic waves. The authors propose an assessment of the shielding by using public frequencies such as TV walkie-talkie signals. The idea is clear, but there are some things to clarify. In my opinion, major revision is required, mainly to better explain some questions.

1) I think it is necessary to state clearly in the introduction that this method is just an assessment, because formal measurement procedure should be done in accordance with standards. Such statement can be found in the Conclusions, but I think a similar one should appear in introduction.

2) Check units of dB and related with it throughout the manuscript. I think sometimes dBm should be used to indicate the signal power, whereas the difference is then in dB. For example, I do not understand what means that “broad frequency is measured as 50 dB”. Or Line 130: “shielding effect is approximately 30 dBm”? Or line 187?

3) Although the idea of the paper is simple and clear some of described measurement procedures are not quite clear. The authors should take into account a reader who is not deep in the subject, but is interested in it. In particular:

- the authors sometimes use mental shortcuts and jargon, which could be not clear for all readers. For example, “measurement of public frequency” - is the frequency measured?

- the measurement procedures should be described more clearly. Maybe figures with the measurement idea could be helpful.

4) Figure 3 has a very low quality. Scale values are almost completely illegible. I know the figures are produced by spectrum analyzer, but maybe they could be given more sharpness. Besides, the caption should be more informative.

5) Table 4 - I think this could figure rather. The caption should be more clear. The scale in the photos is illegible, so maybe it is worth to indicate it in the caption.

6) Unclear fragments:

- Line 129-131: unclear sentence.

- Line 190: “similar strength to the outside of the building”?

- Lines 223-228 - this sentence is very long and I am not sure, if I fully understand it. Consider rearranging it or splitting it.

- Lines 231-236 - what is “direction of the shielding room”?

7) Minor remarks

- Line 81-82: 50 kV/m or 65/m exactly? Doesn’t it depend on distance and maybe bomb power?

- explain all abbreviations when used first, e.g. (ROK, line 36); CI (line 91); (dmB - line 142); RBW, VBW and SWT are explained in lines 225-257, but they appear already in lines 230-231.

- Figure 1: check the numbers in table footer.

- Tables 2 and 3 - strange font used for units,

- Table 2 - unnecessary backslash “\”

- SKT, LG and KT - I guess that they are mobile phone operators in Korea, or maybe I am wrong. It is worth explaining it for a non Korean reader.

Author Response

Response to Reviewers’ Comments

Manuscript ID.: sustainability-1045667

Title: Simplified Measurement Methods of electromagnetic-wave shielding performance using public Frequency

The authors would like to first thank the editor who allowed us opportunities to revise and resubmit the paper. We also sincerely appreciate the anonymous reviewer who provided thorough reviews and valuable comments to help us improve the manuscript. We strongly believe that in the revision we have fully addressed all the reviewer’s comments and concerns and carefully revised the manuscript based on the feedback we have received. Please see the followings below responding to reviewer’s comments and attached file which has highlight with red color on revised part.

Reviewer 2

The paper concerns measurement of shielding performance of building walls against electromagnetic waves. The authors propose an assessment of the shielding by using public frequencies such as TV walkie-talkie signals. The idea is clear, but there are some things to clarify. In my opinion, major revision is required, mainly to better explain some questions.

Comment 1. I think it is necessary to state clearly in the introduction that this method is just an assessment, because formal measurement procedure should be done in accordance with standards. Such statement can be found in the Conclusions, but I think a similar one should appear in introduction.

Response.

We revised some part of the introduction as follows.

Original. (line 73~75)

It is expected that the measurement method of the electromagnetic-wave shielding performance using public frequency proposed in this study will be used in the future for a simplified measurement to determine whether a precise measurement is required.

Revised. (line 78~82)

Since the simple measurement method developed in this study was not adopted as a standard, it can be used as a preliminary investigation to find the vulnerabilities of existing buildings. It is expected that the measurement method of the electromagnetic-wave shielding performance using public frequency proposed in this study will be used in the future for a simplified measurement to determine whether a precise measurement is required.

Comment 2. Check units of dB and related with it throughout the manuscript. I think sometimes dBm should be used to indicate the signal power, whereas the difference is then in dB. For example, I do not understand what means that “broad frequency is measured as 50 dB”. Or Line 130: “shielding effect is approximately 30 dBm”? Or line 187?

Response.

We reviewed all the sentences which are using ‘dB’ or ‘dBm’. Then, we revised some sentence in revised manuscript.

Comment 3. the authors sometimes use mental shortcuts and jargon, which could be not clear for all readers. For example, “measurement of public frequency” - is the frequency measured?

Response.

We changed the title of figure 2 as follow.

Original.

Figure 2. Measurement of public frequency

Revised.

Figure 1. Test places for environment noise measurement

Comment 4. the measurement procedures should be described more clearly. Maybe figures with the measurement idea could be helpful.

Response.

We changed the title and contents of section 4.1 in many part.

Please check the revised manuscript 

Comment 5. Figure 3 has a very low quality. Scale values are almost completely illegible. I know the figures are produced by spectrum analyzer, but maybe they could be given more sharpness. Besides, the caption should be more informative.

Response.

We changed the figure and figure caption as follow

Original. (line 96-97)

(a) Rooftop

(b) Ground

(c) Inside of building

(with window)

(d) Inside of building

(with concrete wall)

(e) Basement

(f) Electromagnetic shield room

Figure 3. Result of simplified EMP shielding test.

Revised. (line 98-99)

(a) Outside(rooftop)

(b) Outside(ground)

(c) Inside of building(nearby window)

(d) Inside of building(nearby concrete wall)

(e) Basement floor

(f) Electromagnetic shield room

Figure 2. Result of environment noise measurement

Comment 6. Table 4 - I think this could figure rather. The caption should be more clear. The scale in the photos is illegible, so maybe it is worth to indicate it in the caption.

Response.

We revised the figure and the caption as follows.

Original.

(a) inside of electromagnetic shielding room

(b) outside of electromagnetic shielding room

Figure 4. measurement result of electromagnetic-wave shielding performance (broadcast signal).

Revised.

(a) outside of electromagnetic shielding room

(b) inside of electromagnetic shielding room

Figure 4. Noise level of mobile communication frequency

Comment 7. Line 129-131: unclear sentence.

Response.

We revised the sentence as follow.

Original.

The Max Hold function was used to measure the maximum instantaneous signal of the data. From the measurements, a maximum of -18.81 dBm in the 880 MHz band was measured under above-ground conditions and -48.18 dBm was measured in the room surrounded by reinforced concrete. From the results, d) the inside of the building (with concrete wall) had a shielding effect of approximately 30 dBm in the 880 MHz band as shown in Figure 3, and the electromagnetic wave shielding tendency can be analyzed.

Revised.

As the result, the maximum environment noises of -27.32 dBm were measured on the rooftop(a), -18.81 dBm on the outside on the ground(b), -23.13 dBm on inside of building nearby window (c), -48.18 dBm on inside of building nearby concrete wall (d), -34.53 dBm on the basement floor (e), and -56.63 dBm on inside of electromagnetic shielding room (f) as shown in the Figure 2.  This result shows that the electromagnetic wave shielding effectiveness of reinforced concrete building is approximately -30 dBm in the 880 MHz band.

Comment 8. Line 190: “similar strength to the outside of the building”?

Response.

We changed the sentence as follows.

Original.

Because a repeater is installed in the building for communication transmission and reception functions inside the building and underground, signals of similar strength to the outside of the building can be received [7].

Revised.

If a repeater is installed in the building for communication transmission and reception functions inside the building and underground, similar strength of signal with the outside of the building can be received [7].

Comment 9. Lines 223-228 - this sentence is very long and I am not sure, if I fully understand it. Consider rearranging it or splitting it.

Response.

We changed the sentence as follows.

Original.

The operating area of the measuring equipment was evaluated by measuring the broadcast/communication signal reception level outside the shielding room and the noise level in the same frequency band inside the shielding room in a test to receive broadcast and communication signals from outside the shielding room using signals of general broadcast and communication frequencies, and determine the difference using shielding performance by receiving signals of the same frequency band inside the shielding room.

Revised.

The signal strength difference between receiving signals on the inside and outside of the shielding room was measured to verify that the broadcast and communication signals could be used for simplified measurement methods of EMP shielding effectiveness. 

Figure 3 shows the shielding room designed for shielding effectiveness test in this study.

Comment 10. Lines 231-236 - what is “direction of the shielding room”?

Response.

We add figure 3 for batter understand.

Comment 11. Line 81-82: 50 kV/m or 65/m exactly? Doesn’t it depend on distance and maybe bomb power?

Response.

there was some mistakes in translation and editing process. The phrase was referenced from reference number 4. So. we revised as follows.

Original. (line 79~82)

 Particularly, the main equipment inside the protection facility should be protected by blocking or sufficiently attenuating the high voltage and current induced by the EMP effect [4]. Generally, the electric intensity of EMPs generated by a nuclear explosion is 50 kV/m or 65 kV/m.

Revised. (line 85~88)

Particularly, the main equipment inside the protection facility should be protected by blocking or sufficiently attenuating the high voltage and current induced by the EMP effect. Generally, the electric field intensity of EMPs generated by a nuclear explosion is from 50 kV/m to 65 kV/m [4].

Comment 12. explain all abbreviations when used first, e.g. (ROK, line 36); CI (line 91); (dmB - line 142); RBW, VBW and SWT are explained in lines 225-257, but they appear already in lines 230-231.

Response.

We change that Things as follow.

Line 36 : ROK -> Republic of Korea

CI in line 91 is miss editing. So, we revised it as follow

C4I(Command, Control, Communications, Computer, and Intelligence)

The detail of RBW, VBW and SWT was in line 254, 256, and 257 even it was mentioned first in line 230. So, the detail is moved to the location where was mentioned first (line 243~244 in revised manuscript)

Comment 13. Figure 1: check the numbers in table footer.

Response.

That was mistakes. we changed it

Comment 14. Tables 2 and 3 - strange font used for units,

Response.

All the units is changed in revised manuscript

Comment 15. Table 2 - unnecessary backslash “\”

Response.

 “\” is deleted

Comment 16. SKT, LG and KT - I guess that they are mobile phone operators in Korea, or maybe I am wrong. It is worth explaining it for a non Korean reader.

Response.

We revised the name of mobile phone operators, and add some description as follow.

Original.

SKT, LG, and KT occupy 49%, 30%, and 21%, respectively, in the Korean mobile communication market.

Revised.

SK telecom, LGU plus, and KT which are mobile phone operators in Korea, occupy 49%, 30%, and 21%, respectively, in the Korean mobile communication market.

Round 2

Reviewer 1 Report

Dear authors,

Congratulations. The new version of the paper is much more complete, and easy for readers to follow.

All previous suggestions were included and new additions were made to the paper.

I agree with the form of the paper.

Author Response

Thanks for your kindly review.

Reviewer 2 Report

Review of sustainability-1045667-peer-review-v2

Simple Measurement Methods of Electromagnetic-wave Shielding Performance for Existing Buildings using Public Frequency Devices

The paper has been improved and I only have two remarks:

1) Some units seem not correct to me in some places.

- lines 117-119: the signal levels 50 dB and 20 dB should be probably 50 dBm (not too high? it’s 100 W) and 20 dBm (?), then the difference is 30 dB.

- Section 4.1.2: dBμV is used in text whereas dBm is visible in Figure 4.

Please check if this is correct.

2) Although Figure 3 has been included in the manuscript, I am not still sure, what the sentence in lines 243-246 means. Exactly, I do not understand “by orientation towards the opposite direction of the shielding room”. Orientation of what? antenna? what is “the direction of the shielding room”? Similarly, “towards the opposite direction of the measuring instrument” is not clear for me.

I am sure that this unclear fragments can be easily improved by the Authors.

Author Response

Response to Reviewers’ Comments

Manuscript ID.: sustainability-1045667

Title: Simplified Measurement Methods of electromagnetic-wave shielding performance using public Frequency

The authors would like to first thank the editor who allowed us opportunities to revise and resubmit the paper. We also sincerely appreciate the anonymous reviewer who provided thorough reviews and valuable comments to help us improve the manuscript. We strongly believe that in the revision we have fully addressed all the reviewer’s comments and concerns and carefully revised the manuscript based on the feedback we have received. Please see the followings below responding to reviewer’s comments and attached file which has highlight with red color on revised part.

Reviewer 2

Simple Measurement Methods of Electromagnetic-wave Shielding Performance for Existing Buildings using Public Frequency Devices

The paper has been improved and I only have two remarks:

1) Some units seem not correct to me in some places.

Comment 1.

- lines 117-119: the signal levels 50 dB and 20 dB should be probably 50 dBm (not too high? it’s 100 W) and 20 dBm (?), then the difference is 30 dB.

Response.

we agree with the comment. It would be mistake. So, we revised the sentence of paper in revised manuscript as follows.

Original.

For example, if the broadcast frequency is measured as 50 dB on the outside of building and 20 dB on the inside of building, it can be deduced that the building itself has 30 dB of electromagnetic shielding effectiveness.

Revised.

For example, if the broadcast frequency is measured as -20 dBm on the outside of building and -50 dBm on the inside of building, it can be deduced that the building itself has 30 dB of electromagnetic shielding effectiveness.

Comment 2.

- Section 4.1.2: dBμV is used in text whereas dBm is visible in Figure 4.

Please check if this is correct.

Response.

We converted the numbers of measured result as follows.

Original.

The maximum signal level for FM broadcasting was 91.53 MHz of 56 dBμV, and was measured as 45 dBμV inside the shielding room, showing a difference of approximately 11 dB. The maximum signal level of terrestrial digital TV broadcasting was 40 dBμV and the frequency in the 183 MHz band was the strongest. Further, the noise level inside the shielding room was measured as 35 dBμV, indicating that there was a margin of approximately 5 dB. The strongest signal was the 880 MHz mobile communication frequency, which was measured as approximately 77 dBμV from the outside and 32 dBμV from the inside, resulting in a difference of approximately 45 dB as shown in Figure 4. This is believed to be a sufficient value to measure the electromagnetic-wave shielding performance of general buildings.

Revised.

The maximum signal level for FM broadcasting was 91.53 MHz of -74 dBm, and was measured as -63 dBm inside the shielding room, showing a difference of approximately 11 dB. The maximum signal level of terrestrial digital TV broadcasting was -67 dBm and the frequency in the 183 MHz band was the strongest. Further, the noise level inside the shielding room was measured as -72 dBm, indicating that there was a margin of approximately 5 dB. The strongest signal was the 880 MHz mobile communication frequency, which was measured as approximately -30 dBm from the outside and -75 dBm from the inside, resulting in a difference of approximately 45 dB as shown in Figure 4. This is believed to be a sufficient value to measure the electromagnetic-wave shielding performance of general buildings.

Comment 3.

2) Although Figure 3 has been included in the manuscript, I am not still sure, what the sentence in lines 243-246 means. Exactly, I do not understand “by orientation towards the opposite direction of the shielding room”. Orientation of what? antenna? what is “the direction of the shielding room”? Similarly, “towards the opposite direction of the measuring instrument” is not clear for me.

Response.

We agree that the sentence is not clear. It was because of unclear writing on original and mistake on translation. So, we revise the sentence as follows.

Original.

The test was conducted by orientation towards the opposite direction of the shielding room at the center of the test room when measuring the outside of the shielding room, and towards the opposite direction of the measuring instrument at the center of the shielding room when measuring the inside of the shielding room.

Revised.

The antenna was conducted by orientation towards to the opposite direction from the shielding room when measuring the outside of the shielding room, and towards to outside at the center of the shielding room when measuring the inside of the shielding room.